# New measures of deep soil water recharge during vegetation restoration process in semi-arid regions of northern China

Yiben Cheng[1,5*], Xinle Li[2*], Yunqi Wang[1,5], Hongbin Zhan[3], Wenbin Yang[4], Qunou Jiang[1,5]

[1]School of Soil and Water Conservation, Beijing Forestry University, Beijing 100083, China

[2]Experimental Center of Desert Forestry, Chinese Academy of Forestry, Dengkou 015200, China

[3]Department of Geology & Geophysics, Texas A&M University, College Station, TX 77843-3115, USA.

[4]Institute of desertification control, Chinese Academy of Forestry, Beijing, 100093, China

[5]Jinyun Forest Ecosystem Research Station, School of Soil and Water Conservation, Beijing Forestry University, Beijing 100083, China

Corresponding Author:

Yiben Cheng (chengyiben@bjfu.edu.cn), Xinle Li (nxylxl@126.com).

**Abstract**

Desertification in semi-arid regions is currently a global environmental and societal problem. This research attempts to understand whether a 40-year-old rain-feed Artamisia sphaerocephala Krasch sand-fixing land in Three North Shelterbelt Program (3NSP) of China can be developed sustainably or not, using a newly designed lysimeter to monitor the precipitation-induced deep soil recharge (DSR) at 220 cm depth. Evapotranspiration is calculated through a water balance equation when precipitation and soil moisture data are collected. Comparison of soil particle sizes and soil moisture distributions in artificial sand-fixing land and neighboring bare land is made to assess the impact of sand-fixing reforestation. Results show that such a sand-fixing reforestation results in a root system being mainly developed in the horizontal direction and the changed soil particle distribution. Specifically, the sandy soil with 50.53% medium sand has been transformed into a sandy soil with 68.53% fine sand. Within the Artamisia sphaerocephala Krasch sand-fixing experimental area, the DSR values in bare sand plot and Artamisia sphaerocephala Krasch plot are respectively 283.6 mm and 90.6 mm in wet years, reflecting a difference of more than three times. The deep soil layer moisture in semi-arid sandy land is largely replenished by precipitation-induced infiltration. The DSR values of bare sandy land plot and Artamisia sphaerocephala Krasch plot are respectively 51.6 mm and 2 mm in dry years, a difference of more than 25 times. The proportions of DSR reduced by Artamisia sphaerocephala Krasch is 68.06% and 96.12% in wet and dry years, respectively. This research shows that Artamisia sphaerocephala Krasch in semi-arid region can continue to grow and has the capacity of fixing sand. It consumes a large amount of precipitated water, and reduces the amount of DSR considerably.

**Keywords:** Semi-arid land, Artamisia sphaerocephala Krasch, rain-feed vegetation,

45  replantation, infiltration, 3NSP

## 1. Introduction

Desertification is currently a global environmental and societal concern (Reynolds et al., 2007b). Arid region covers about 41% of the Earth's surface, and supports more than 38% of the world's population. 20% of these areas have experienced serious land degradation, which is expected to affect the survival of 250 million people (Reynolds et al., 2007a;Dregne and Chou, 1992;D'Odorico et al., 2013). In 1992, the United Nations adopted the International Convention to Combat Desertification in order to focus on desertification issues (Bestelmeyer et al., 2015). With no exception, China is also facing severe desertification problems (Liu and Diamond, 2005). Up to 2010, the total desertification area in China is 2,623,700 $km^2$, which is 27.33% of the country's entire land area. Among this, the arid region desertification area is 1,158,600 $km^2$ (44.16% of the total desertification area of China), the semi-arid region desertification area is 971,600 $km^2$ (37.03% of the total desertification area of China), and the sub-humid arid region desertification area is 493,500 $km^2$ (18.81% of the total desertification area of China). To battle desertification, an effective prevention and control measure is to build shelterbelts, using artificial sand-fixing vegetation (Tao, 2014).

The Three North Shelterbelt Program (3NSP), a reforestation program initiated in 1978 in Northeast, Northwest, and North China is the largest shelterbelt project in China (Wang et al., 2004;Wang et al., 2010b). It has been constructed for 40 years, and plays a key role for desertification prevention in Northeast, Northwest, and North China (Li et al., 2004). The shelterbelts of 3NSP have slowed down, halted, and even reversed the desertification process in Northern China (Zha and Gao, 1997;Wang et al., 2012). According to NASA's latest observations, the restoration of vegetation has shown some

signs of reversing the trend of desertification in China, accounting for a quarter of the

Earth's new green areas (Chen et al., 2019).

It is unquestionable that the implementation of 3NSP in China has reduced aeolian

erosion, and improved the overall living environment in the impacted regions (Hanjie

and Hao, 2003). However, it is undeniable that the poor choices of vegetation species

in some areas of 3NSP has resulted in consumption of a large amount of water resources,

causing shortage of water supply to meet other needs, thus threatening the sustainable

development of the regions (Wang et al., 2010a). Furthermore, a high planting density

in some areas resulted in the death and/or malfunction of a large number of trees (Duan

et al., 2011). In contrast, shrubs and herb sand-fixing vegetation appear to grow

healthily, thus receive great interests to become proper choices of vegetation species

for desertification prevention (Tao, 2014). The infiltration process also closely related

to the development of plant roots, the distribution depth and development direction of

roots in the soil are related to precipitation infiltration(Fan et al., 2017).

To understand the impact of afforestation to the ecohydrological system, thus to

assess the long-term sustainability, especially after vegetation reconstruction(Cheng et

al., 2020). Soil water is the most important factor in this system, thus we need to know

how the soil moisture changes in these area(Cheng et al., 2020). Evapotranspiration(ET)

is also an important ecological indicator in semi-arid regions, the methods of directly

measuring ET include Lysimeter, Eddy correlation method, Bowen ratio method, Large

aperture scintillation method, etc (Billesbach and Meteorology, 2011;Maes et al., 2019).

Taking the most advanced Eddy correlation method as an example, the measurement

error may be 20% or higher and the required monitoring conditions are quite demanding

(Burba and Anderson, 2010). Furthermore, it is difficult to avoid the influence of human

factors on the experimental results. In this research, we try to solve these problems, we

select a typical semi-arid area in 3NSP for this research. *Artamisia sphaerocephala Krasch* (ASK) is a unique Chinese native sand-fixing shrub plant with strong adaptability (Wang et al., 2013). ASK sand-fixing land developed on top of bare sandy land has increased evapotranspiration. Meanwhile, because of the form of an organic-rick biofilm commonly seen in ASK forest, the near surface soil permeability has been reduced (Su and lin Zhao, 2003). This will reduce the soil infiltration capacity, resulting in the concentration of soil moisture in shallow soils, and reducing the replenishment of soil moisture in deep soils. In order to understand the soil moisture variation and deep soil recharge (DSR) changes resulted from ASK sand-fixing forest, this research choose a 40-year-old ASK sand-fixing land as the experimental site and use a new designed Lysimeter directly measure the DSR. We have also conducted a comparative research using a bare sandy land 400 m away from the ASK sand-fixing land site. This research focuses on monitoring precipitation-induced infiltration, soil moisture distribution, and DSR changes in ASK sand-fixing land, then calculate the annual change of ET on a small scale according to the principle of water balance. We try to answer the following questions through in-situ measured result: 1) Does the vegetation reconstruction blocked the mutual conversion of atmospheric water, surface water, soil water, and groundwater? 2) How much influence does vegetation reconstruction have on precipitation infiltration and DSR? 3) Is this kind of vegetation reconstruction sustainable in north China?

## 2. Material and Methods

### 2.1 Research area description

Figure 1 shows the research site which is located in Ejin Horo Banner, on the Eastern margin of Mu Us Sandy Land in the Ordos basin of China, with a geographic location of 39°05'02.8 N, 109°35'37.9 E, and an altitude of 1303 m above mean sea

level (m.s.l.). The groundwater table between sand dunes are 5.3-6.8 m below ground surface. The climate is within the semi-arid continental monsoon climate zone. Annual precipitation concentrates from July to September and is highly sporadic. The average annual precipitation from 1960 to 2010 is 358.2 mm (Li et al., 2009). The average annual temperature of this area is 6.5℃, with about 151 days of frost-free season, and the lowest temperature is -31.4 ℃. The average annual potential evapotranspiration is 1809 mm, the average annual sunshine is 2900 hours, and the average annual wind speed is 3.24 m/s. The research area is located in relatively gentle mobile dunes, and the soil type is aeolian sandy soil (Liu et al., 2015).

The experimental site was flat sandy land before ASK was planted for sand control 40 years ago. After 40 years of development, the region is dominated by ASK, scattered Rhamnus parvifolia, Chenopodium glaucum, Setaria viridis and the field average vegetation coverage has reached 80%. The site is relatively homogeneous that brings convenience to experiment so we can use one point experimental observation result to represent the entire homogeneous site.

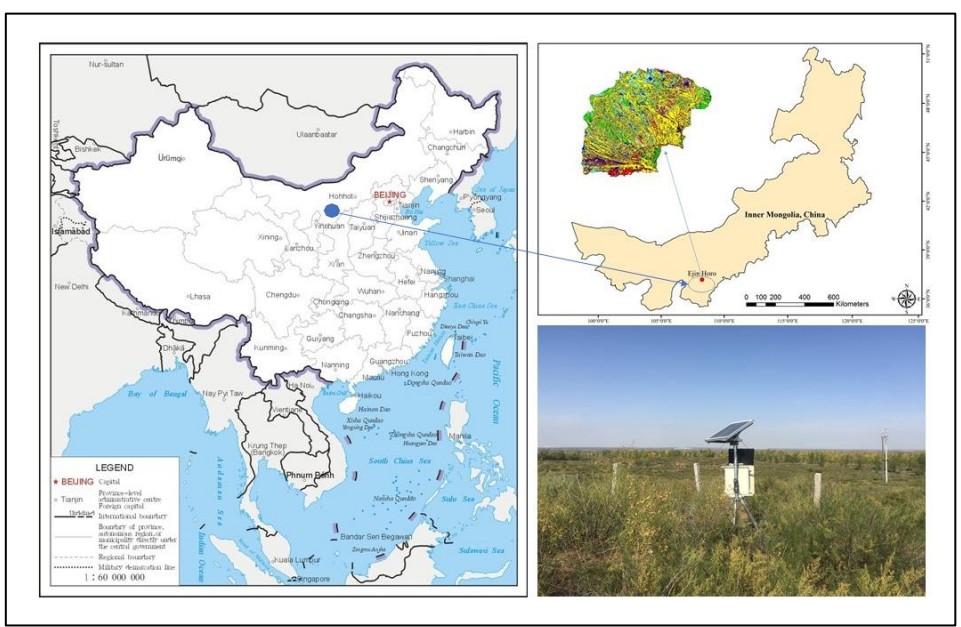

Figure 1. Overlook of the experimental field.

## 2.2 Experimental design

### 2.2.1 Root system distribution survey, soil moisture and DS monitoring

This research chooses five ASK plants with similar heights and crown widths, in which the heights are around 60 cm above the ground. Using the whole root system excavation method, the plant soil is excavated layer by layer with a 20 cm vertical interval, until there are no observable roots. As the deepest root is at a depth of 120 cm (the root system will be discussed in details later in this section), thus the deepest soil moisture that the plant can utilize is 180 cm (120 cm root depth plus 60 cm capillary rise, where the capillary rise is calculated based on the soil texture from experimental plot) (Cheng et al., 2017). The 180 cm depth can be regarded as the maximal depth of evapotranspiration. A new lysimeter is used to measure the deep infiltration, or deep soil recharge (ds) at a depth of 220 cm (to avoid root water absorption), 40 cm below the maximal depth of surface evapotranspiration. The newly designed lysimeter is improved on the basis of the traditional lysimeter, but it has a reduced size and a new water balance part to improve the measurement accuracy. As shown in the Figure 2, the measurement surface is transferred from the soil surface to soil layer at any designated depth. The detailed explanation of such a lysimeter has been documented in a previous research of Cheng et al. (2017) and will not be repeated here. To understand the soil condition in the research site, the sandy soil samples are collected using a ring cut method, layer by layer with a 20 cm vertical interval, until reaching a 220 cm depth. Soil samples from five ASK plots were collected and mixed together, soil particle size distribution measurements are conducted using a laser particle size analyzer (Mastersizer 2000, Salver, U.K.). We use EC-5 soil moisture probe to measure every 20 cm soil layer of the first 100 cm depth, and every 40 cm soil after the first 100 cm depth until reaching 220 cm depth. The reason of doing so is because the shallow soil

layer has roots thus is monitored more closely while the deep soil is relatively uniform and has less roots, thus can be monitored more sparsely.

To study the soil water dynamics of ASK plot, we selected a typical ASK plot in the Mu Us Sandy Land and an adjacent bare sandy plot as a comparison study to quantify the differences in the characteristics of soil water dynamics in bare sandy plot and ASK plot. The experimental design is shown in Fig.2, Fig.3 and explained sequentially as follows.

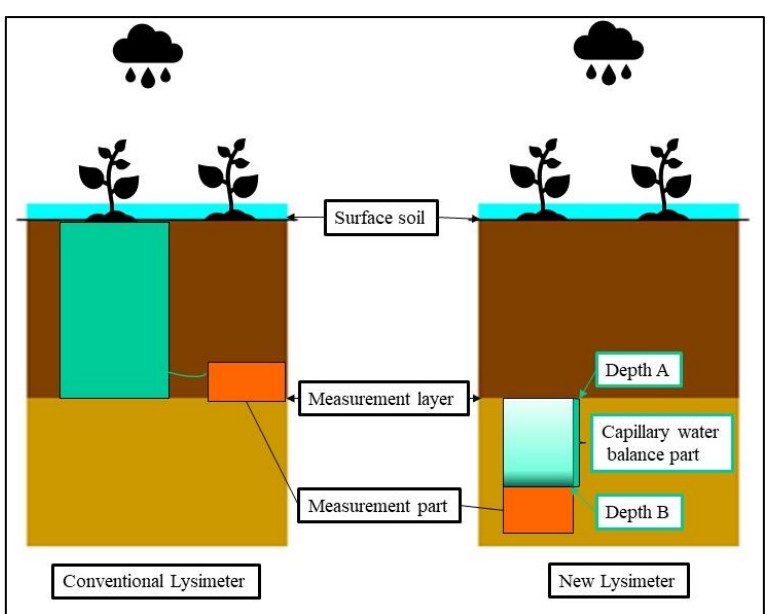

Figure 2 Schematic diagram of conventional Lysimeter and new designed Lysimeter.

The conventional Lysimeter uses an impermeable container (constructed all the way from ground surface downward) to wrap the soil column, blocking the horizontal flow of the soil layer in the root zones. Meanwhile, if a conventional Lysimeter is used, the vegetation needs to be transplanted into the container, so the soil structure and the vegetation root system will be disturbed. If the roots of the vegetation are too long, it is impossible to measure the DSR with a conventional Lysimeter. The new Lysimeter has an upper water balance part and a lower measurement part which can directly measure the water flux (Cheng et al., 2017b;Cheng et al., 2018b). The height of the balance part

is equal to the capillary water rising height of this sandy soil. After irrigation and

standing still, the balance part will reaches a balanced state that the soil at the bottom

is saturated, and the top is the highest distance that the capillary water can rise.

Specifically, the flux infiltrating into the balance part at the depth of the measurement

face should equal the flux exiting the balance part and entering the measurement part.

There is no need to build an impervious container to wrap the vegetation tested for the

new Lysimeter above the measurement face. The plot is relatively homogeneous, so we

can use one point experimental observation result to represent the entire homogeneous

field. The conventional lysimeter can measures the DSR of a plant, but the new

lysimeter measures the DSR of a certain soil layer. For a relatively uniform plot, this

measured DSR would be more accurate.

In order to minimize the disturbance of the original soil structure, we need to water

both plots in advance before installing the instruments. Firstly, watering the soil in the

test area makes the relatively dry sandy soil stable and easy to excavate, as the native

dry sandy soil is relatively loose. Secondly, after watering the ASK plot, we start to

excavate a soil profile vertically downward at a distance of 1 meter from the main

branch of ASK, reaching a depth of 3.2 meters. After this, at the depth of 3.2 meters,

we excavate horizontally toward the location of the main branch of ASK to a distance

about 1.3 m. Eventually, a body with a height of 1 m, a length and width both of 0.3 m

is excavated to install the lysimeter right below the main branch of ASK. By doing so,

the distance from the ground surface to the top of lysimeter is 220 cm, and the root

system (which is less than 220 cm deep) will not be disturbed. Meanwhile, as the plot

has been watered to make the soil stable, no collapse of soil has occurred during the

installation of the lysimeter. Thirdly, after putting the lysimeter in place, we use in-situ

soil to backfill. During this process, one needs to continuously water each layer of

backfill to ensure that the soil is relatively compact. For the installation of lysimeter in the bare sand plot, it is straightforward as one does not need to worry about the disturbance of integrity of the root system. For such a plot, as shown in Fig.3 one can water the soil first, then excavate a square of 1 meter by 0.3 meters to a depth of 3.2 meters to install the lysimeter. After the installation of lysimeter, one can backfill using

native soil, making sure to continuously water each layer of backfill to ensure the soil compaction. Soil moisture probes are installed at different depths for both plots. Finally, wait for the watered plots to stabilize to its pre-excavation status, since pre-watered sandy plot and excavated sand layer will take six months to settle down and meet the requirements of the experiment. Then one can start the experiments. There is a notable

limitation of this new lysimeter that should be improved in the future. When measuring DSR, a gauge with a measurement accuracy of 0.2 mm was used to automatically record the amount of DSR. The measuring mechanism of this gauge was that when the accumulated amount of DSR reached a certain amount (0.2 mm), which is the downward volumetric flux over a unit area over a certain time lapse, then a data point

will be recorded. When the amount of DSR is large or there is continuous DSR, there will be no measurement problem, but when DSR is very small (less than 0.2mm), it is impossible to know precisely the DSR variation over that time lapse. In the future, we need a more sensitive measuring apparatus that can precisely record the DSR variation with time in a higher precision.

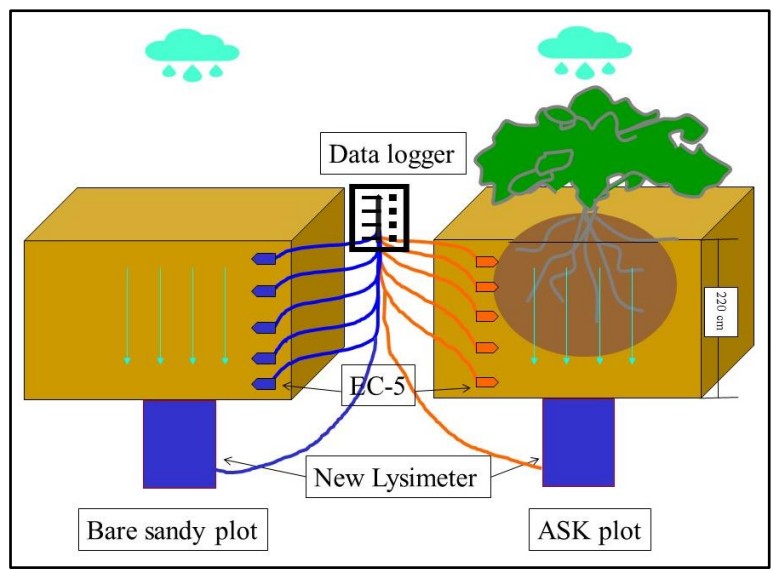

Figure 3. Design of Precipitation-DSR observation site

### 2.2.2 Water balance of rain-fed ASK forest land

When precipitation reaches ground surface in semi-arid sandy land, the infiltration rate is usually unpredictable, it may evaporate or run away, or infiltrate. Years of observation records in the area show no occurrence of surface runoff (Cheng et al., 2021). The water infiltrating into the soil goes through a redistribution process. Part of it is absorbed and utilized by plants' root system, and part of it is stored in soils as soil moisture. The rest will infiltrate passing the maximal depth of evapotranspiration depth and eventually recharges the groundwater system. This research uses the following water balance method to calculate moisture distribution at different depths:

$$P + C_m*d - DSR - E = \pm \triangle W \tag{1}$$

where $P$ is annual precipitation (mm) measured by a rain gauge as the volume per unit square meters, $C_m$ is soil volumetric moisture content ($m^3/m^3$), $d$ is soil column depth to be measured (mm), DSR is annual deep soil recharge (mm), measured by the newly designed lysimeter as the volume per unit square meters, $E$ is annual evapotranspiration (mm) which is the volume of water lost to the atmosphere due to

evapotranspiration per square meters, and $\triangle$W is the annual soil moisture storage

change per unit square meters (mm).


### 3. Results

### 3.1 Root system distribution

This research selects representative plants and excavate the soil profile to research

the ASK root system growth range. The results show that the ASK root system

distribution is umbrella-shaped, as shown in Table 1. The root system distribution range

mainly concentrates within 0-60 cm depth, and can reach as deep as 120 cm. The main

root grows through the entire depth. The lateral roots are distributed around the main

root and can reach a 200 cm diameter horizontally. The density of lateral roots gradually

decreases when moving away from the central main root. The lateral roots mainly

concentrate within depths of 20-60 cm. From ground surface to a depth of 40 cm, the

root system gradually increases, and reaches the maximum density at the 40 cm depth.

The dry weight of root between 20-40 cm layer is 51.77% of the weight of the entire

root system. The root system gradually decreases after depths of 40 cm, with the deepest

root system depth of 120 cm. The results show that the ASK root system in this area is

mainly developed in the horizontal direction, which confirms that rainfall is the main

water supply for plants in the Mu Us Sandy Land. This conclusion is based on the

following reasons. The root development of plans is closely dependent on the source of

water supply for the root system, and there are generally two sources of supply: a)

rainfall-induced downward infiltration and b) uptake of groundwater from the

underneath soil and aquifer. If the primary source of supply for the ASK root system

comes from the deep groundwater table, then the root prefers to grow vertically in order

to access the underneath groundwater. On the other hand, if the primary source of

supply for the root system comes from the rainfall-induced infiltration, the root system prefers to grow horizontally to maximize the intercept of such infiltrated water, and the field observation results confirm that this is the case in Mu Us Sandy Land.

Table 1. Root distribution of the ASK in the vertical direction

| Excavation depth(cm) | Root dry matter content (g) | The dry matter accounts for the weight ratio of the whole root system (%) |
|---|---|---|
| 0-20 | 21.85 | 13.26 |
| 20-40 | 85.32 | 51.77 |
| 40-60 | 30.56 | 18.54 |
| 60-80 | 14.86 | 9.02 |
| 80-100 | 8.57 | 5.2 |
| 100-120 | 3.64 | 2.21 |

## 3.2    Effect of ASK on soil development

There are many factors that affect the soil particle size, including soil crust, vegetation root secrete acidic substances to decompose the parent material, ionic
strength, flow rate and surface vegetation fixed sand dust (Yan et al., 2013;Yu et al., 2013;Zhang et al., 2011). The soil particle size of each layer is also different. It is necessary to analyze each soil layer one by one and it is not easy to see the main affecting factors. In this research, to understand the impact of ASK on the local soil, the ASK soil samples and bare land soil samples are collected and sorted based on U.S.
Department of Agriculture's soil particle size grading scheme, we collected samples of every 20 cm depth and mixed them together, treated the entire 220 cm thick soil layer each as a homogenized system.

The soil particle size distribution was measured using the MS2000 soil particle size analyzer produced by Malvern, UK. Samples need to be pretreated before the

experiment. All soil samples have passed through a 2 mm soil sieve, added 30% $H_2O_2$

solution to remove organic matter (including biological crust) from the sample, then

add NaHMP solution to fully dissolved, and shake 30 seconds to destroy the

microaggregate structure of the soil particles.

Table 2 shows the particle size distributions in both the ASK plot and bare sandy

plot. Overall, in ASK plot, the medium sand is 19.26%, the fine sand is 68.53%, the

very fine sand (or powder sand) is 9.35%, and silt is 2.86%. The soil particle size

distributions of the bare sandy plot are as follows. The coarse sand is 3.23%, the

medium sand is 50.53%, the fine sand is 36.06%, the very fine soil is 7.19%, and the

silt is 2.99%. Comparing the results in ASK plot and bare sandy plot, one can see that

the main soil type in the ASK plot is fine sand (68.53%), and the main soil types in bare

sandy plot is medium (50.53%) and fine sands (36.06%). Another notable point is that

there is 3.23% of coarse sand in the bare sandy plot, but no coarse sand in the ASK plot.

There are clear evidences that the sand-fixing vegetation changes the particle size

distribution of the soil (Fearnehough et al., 1998;Pei et al., 2008). A few possible

reasons may be responsible for such a change. First, the fine-sand in the 220 cm-thick

soil of the bare sandy land is easily removed or eroded from its original position under

the force of wind, which initiates sand movement both horizontally and vertically (as

suspended particles carrying away by wind), consequently the content of fine sand in

the bare sandy land decreases, and the soil structure continuously coarsens. In contrast

to this, the content of fine particle in the ASK plot is significantly higher than that in

the bare sand. This is largely due to the presence of vegetation in the ASK plot which

has substantially subdued the eroding force of wind. In another word, ASK essentially

protects the fine sands in the soil to be removed or eroded by wind force. This

observation is direct evidence showing that vegetation has a positive role in improving

soil particle size composition by maintaining the fine sand particles in the plot. However, one must also be aware that such a change of particle size distribution is a consequence of a complex interplay of aerodynamic force, sand mass movement mechanics, and root-soil interaction force, which are not completely understood up to now and needs further investigation. In summary, the sand-fixing vegetation in northern China not only fixes the quicksand, but also greatly improves the soil texture, which is far faster than traditional expectations.

Table 2. The distribution of soil particle size in research site

| Particle size distribution | Extra coarse sand | Coarse sand | Middle sand | Find sand | Very fine sand | Silt sand |
|---|---|---|---|---|---|---|
| Diameter range (mm) | >1.0 | 1.0-0.5 | 0.5-0.25 | 0.25-0.1 | 0.1-0.05 | <0.05 |
| ASK plot | 0.00 | 0.00 | 19.26 | 68.53 | 9.35 | 2.86 |
| Bare sandy plot | 0.00 | 3.23 | 50.53 | 36.06 | 7.19 | 2.99 |

### 3.3    Annual soil moisture variation of rainfed ASK plot

The experimental area is located in Northern China, with more than three months of intermittent frozen soil period in winter. Multi-year observations show that the frozen soil period is from December of the previous year to March of the following year. The annual soil moisture variation in 2015 is shown in Figure 3. According to the change of soil volumetric water content and the influence of precipitation, the whole year is divided into five stages, which are the thawing recharge period, germination consumption period, rain season recharge period and plant dormancy period (the freeze-thaw period refers to the top soil (2 meters depth) from the beginning of freezing to the complete melting of the frozen soil. The germination period begins from the end of freeze-thaw period to the period when branches of ASK are enlarged and one or two

new leaves start to grow. The rainy season refers to a period of relatively concentrated

precipitation experienced after the germination period of ASK plot in this region. the

frozen soil period is not shown in Figure 4). From 2015 to 2018, the trend of soil

moisture is basically the same, only the time of the rainy season and the amount of

annual precipitation are different.

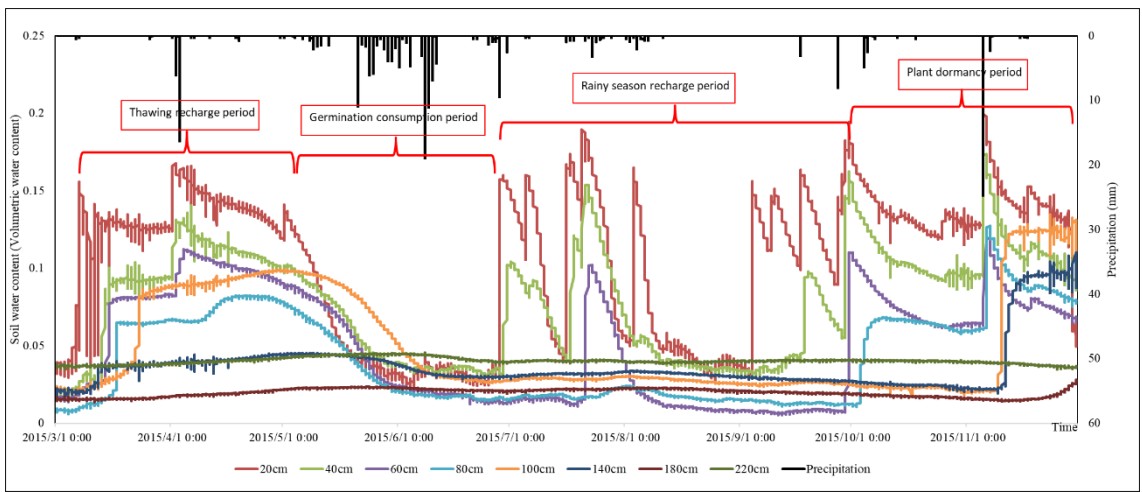

Figure 4 Daily soil water content distribution of ASK plot in 2015.

After March 6th, the melting snow in ground surface leads to increased soil moisture

contents. Around this time, ASK is still in winter dormancy, and does not absorb soil

moisture. As shown in Figure 4, from March 6th to May 5th, soil moisture increases

significantly. Soil moisture resulted from melting snow can infiltrate into depths of 100

cm to 140 cm. After April 25th, ASK starts germination, and soil moisture gradually

decreases. From April 25th to June 27th, there are 31 observed precipitation events in

total. The maximum precipitation is 18.8 mm, and the minimum precipitation is 0.2

mm. However, these precipitation events did not change the decreasing trend of soil

moisture. This means that during the germination and early growth periods, the

moisture absorbing capacity of the ASK root system is extremely high. There is a 9.4

mm precipitation event on June 28th, and the infiltration associated with this event can

reach a depth of 20 cm. This means that the growth of ASK starts to slow down around

this time, and the shallow soil moisture starts to increase. In October, temperature drops and ASK starts to enter winter dormancy. There is a 4.2 mm precipitation event on

October 4[th], and the infiltration associated with this event can reach a depth of 60 cm. There is a 24.6 mm precipitation event on November 7[th], and the infiltration associated with this event can reach a depth of 140 cm. Soil moisture at 220 cm depth changes very mildly. The results show that though DSR occurs in all seasons, especially during freeze-thaw period, due to vegetation consumption, the amount of DSR is relatively

small.

### 3.4     Effects of annual precipitation on soil moisture and DSR

### 3.4.1   Comparison of DSR on rain-feed ASK land and bare sandy land

For deep soil moisture variation and distribution, this research uses a newly designed lysimeter to measure DSR on-site (Cheng et al., 2017). The soil layer may be

disturbed after the instrument is installed in 2015, so the 2015 precipitation-infiltration data are not used in this study. Results are shown in Table 3. From 2016-2018, the precipitations of bare sandy land are 464.8 mm, 313.4 mm, 245.2 mm, and DSR are 283.6 mm, 67.6 mm, 51.6 mm, respectively. The ratios of DSR to annual precipitation are 60.02%, 21.57%, 21.04%, respectively. The experimental plot of Artamisia is less

than 100 m away from the bare sandy land plot, the annual precipitation is basically the same, and DSR values are 90.6 mm, 31.2 mm, 2 mm, respectively. The ratios of DSR to annual precipitation are 19.49%, 9.96%, 0.82%, respectively. According to above data, DSR of the bare sandy land is obviously higher than the Artamisia plot. On Artamisia plot, the interception of the aboveground vegetation, root absorption,

evapotranspiration consumes a large amount of water resources, which affects the production of DSR.

Table 3. Comparative of precipitation and DSR in ASK land and bare sand field

| Year | Field type | Precipitation (mm) | DSR (mm) | D/P (%) |
|------|-----------|--------------------|----------|---------|
| 2016 | Bare sand plot | 464.8 | 283.6 | 60.02 |
|      | ASK plot | 464.8 | 90.6 | 19.49 |
| 2017 | Bare sand plot | 313.4 | 67.6 | 21.57 |
|      | ASK plot | 313.4 | 31.2 | 9.96 |
| 2018 | Bare sand plot | 245.2 | 51.6 | 21.04 |
|      | ASK plot | 245.2 | 2 | 0.82 |

When the annual precipitation at a particular year is higher than 358.2 mm, it is considered a wet year; if the annual precipitation at a particular year is lower than 358.2 mm, it is considered a dry year. As shown in Table 3, 2016 is a wet year, 2017 is a normal year, and 2018 is a dry year. In the wet year, the deep soil moistures of the two experimental sites were greatly supplemented, and the effect of bare sand was more obvious. The amount of DSR in the dry years is significantly reduced on both plots, especially in the Artamisia plot, from 90.6 mm in wet years to 2 mm in dry year. Based on these, one can conclude that in semi-arid areas, though vegetation cover can fix mobile sand dunes, it consumes a lot of water resource. Bare sandy land can transport large amounts of water resource to shallow groundwater.

### 3.4.2 Precipitation response to soil moisture and DSR on two experiment plots

The relationship between precipitation and soil moisture content fluctuations and DSR in 2016 is shown in Figure 5 (ABCD). There are 84 precipitation events throughout the year in 2016, with the maximum precipitation amount of 137.2 mm/d happened on July 10th. According to local weather station data, this is the second largest daily precipitation since 1950, and the minimum precipitation amount of 0.2 mm/d happened 11 times throughout the year. On bare sandy land, there are 138 infiltration

events throughout the year in 2016, with the maximum DSR amount of 24 mm/d, happened on July 11[th], and the minimum DSR amount of 0.2 mm/d happened 25 times throughout the year. On ASK plot, there are 42 infiltration events throughout the year in 2016, with the maximum DSR amount of 27 mm/d, happened on July 13[th], and the minimum DSR amount of 0.2 mm/d happened 22 times throughout the year. The comparison of these two sets of DSR data shows that ASK can substantially reduce the soil moisture infiltration, DSR of Artamisia plot is reduced by 68.05% compared to bare sandy plot. Heavy precipitation completely wets the entire soil layer and forming a moisture transport channel that facilitates the transport of moisture throughout the soil layer. In bare sandy land, as the entire soil layer is wet, the subsequent small precipitation can also replenish the deep soil layer moisture, as shown in Figure 3A. In the experimental area of Artamisia plot, heavy rainfall wets the entire soil layer, but for the root system soil water consumption, the subsequent small precipitation cannot significantly replenish the deep soil moisture, as shown in Figure 4D.

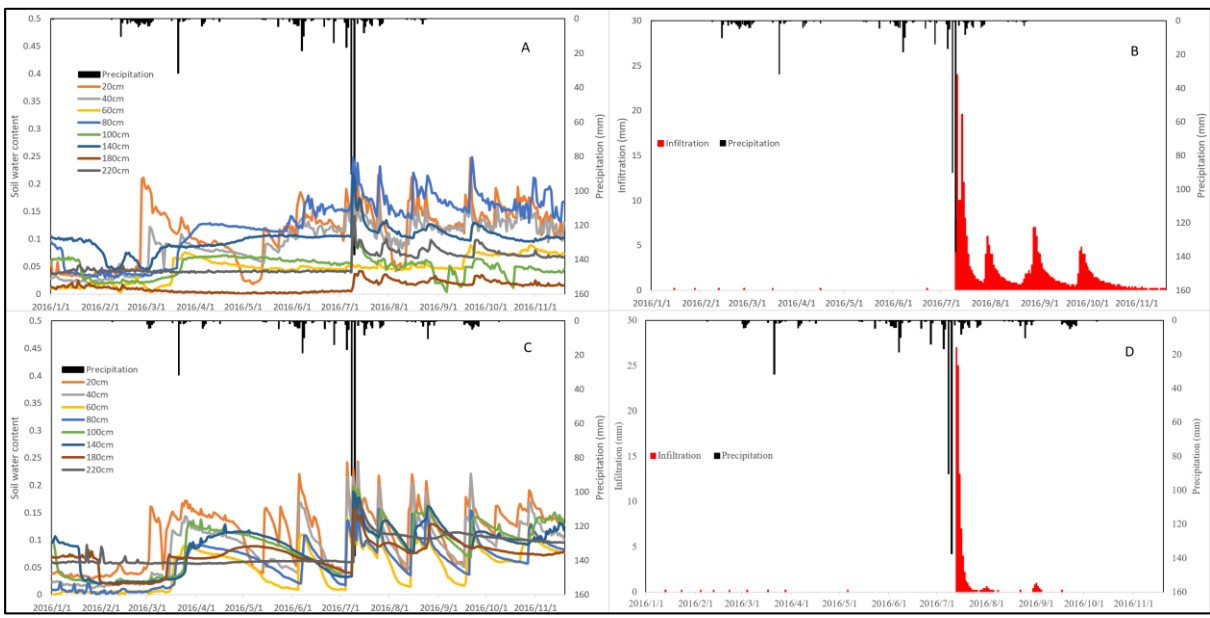

Figure 5. Effects of precipitation on soil moisture (A) and DSR (B) in bare sandy land plot; Effects of precipitation on soil moisture (C) and DSR (D) on ASK plot, 2016.

The relationship of precipitation on soil moisture and DSR in bare sandy land plot and ASK plot of 2017 is showed in Figure 6(A-D). There are 94 precipitation events throughout the year in 2017, with the maximum precipitation amount of 18.8 mm/d happened on June 29th, and the minimum precipitation amount of 0.2 mm/d happened 24 times throughout the year. On bare sandy land, there are 178 infiltration events throughout the year in 2017, with the maximum DSR amount of 8 mm/d, happened on August 23th, and the minimum DSR amount of 0.2 mm/d happened 128 times throughout the year. On the ASK plot, there are 52 infiltration events throughout the year in 2017, with the maximum DSR amount of 1.6 mm/d, happened on September 5th, and the minimum DSR amount of 0.2 mm/d happened 21 times throughout the year. There were only 6 times of infiltration in bare sand plot from January to April in 2016, and 50 times in 2017, as shown in Figures 4 and 5. Since the surface is frozen at this time, there will be no surface infiltration. The source of infiltration in the first three months is essentially from the soil layer reservoir of 2016. One can speculate that the accumulation of water in the soil in the previous year can continue to infiltrate to the second year. This also makes it difficult to subdivide which precipitation process induced how much soil water content. In 2017, there was less precipitation than the previous year, so the DSR was reduced in both plots, especially the ASK plot. Only after the vegetation had dried up in September 9th did a large infiltration process occurred.

The results show that in the Mu Us Sandy Land, whether there is vegetation coverage or not, DSR occurs in all seasons of the year and there is a significant difference in terms of DSR characteristics in the bare sand plot and the ASK plot. More specifically, the annual DSR of the bare sandy lands reaches 3.13 times of that of the

ASK land. After the freeze-thaw period, the ASK root system begins to utilize the soil

moisture, and soil moisture consequently decreases significantly.

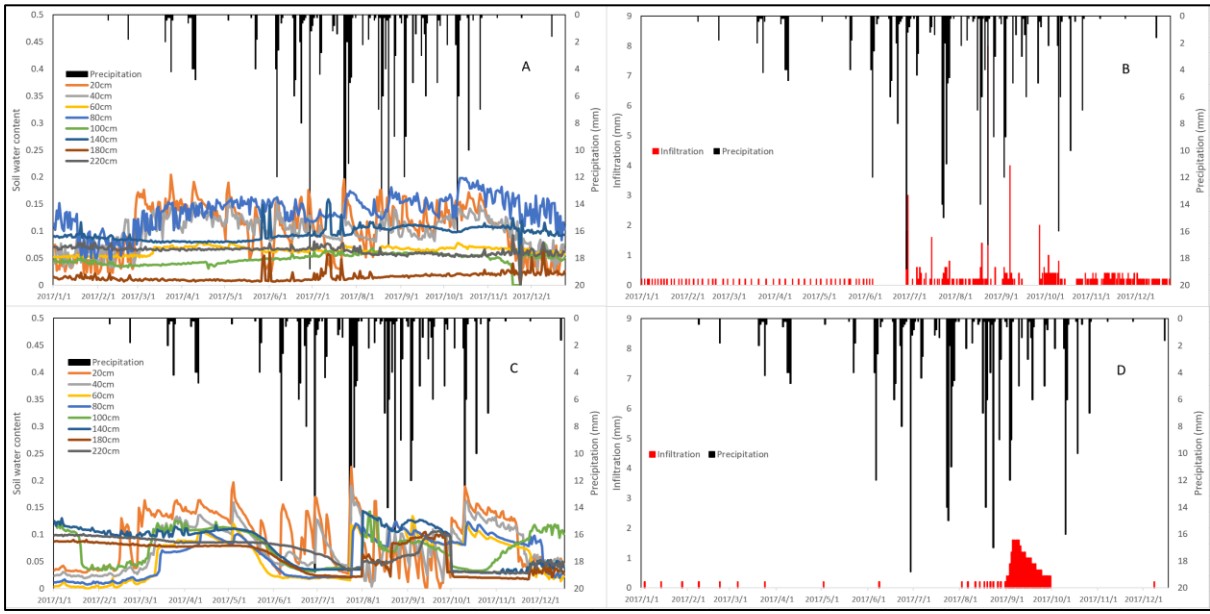

Figure 6 Effects of precipitation on soil moisture (A) and DSR (B) in bare sandy land plot; Effects of rainfall on
soil moisture (C) and DSR (D) on ASK plot, 2017.

The relationship between precipitation and soil moisture content fluctuations and

DSR in 2018 is shown in Figure 7(A-D). There are 71 precipitation events throughout

the year in 2018, with the maximum precipitation amount of 30 mm/d happened on

August 31th, and the minimum precipitation amount of 0.2 mm/d happened 15 times

throughout the year. On bare sandy land, there are 122 infiltration events throughout

the year in 2018, with the maximum DSR of 1.6 mm/d happened on June 4th, and the

minimum DSR of 0.2 mm/d happened 74 times throughout the year. On ASK plot, there

are 10 infiltration events throughout the year in 2016, with the maximum and the

minimum DSR at the same amount of 0.2 mm/d happened 10 times throughout the year.

The results show that under the heavy precipitation event on August 31th, 2018,

DSR in the bare sandy land is obviously visible. The precipitation replenishes deep soil

layer and shallow groundwater. However, in the ASK plot, a large percentage of

precipitation-induced infiltration is intercepted by vegetation coverage, meaning that

the sand-fixing vegetation strongly affects the infiltration process and has a greater

impact on groundwater recharge. At the same time, DSR can be found in both plots in

all seasons throughout the year.

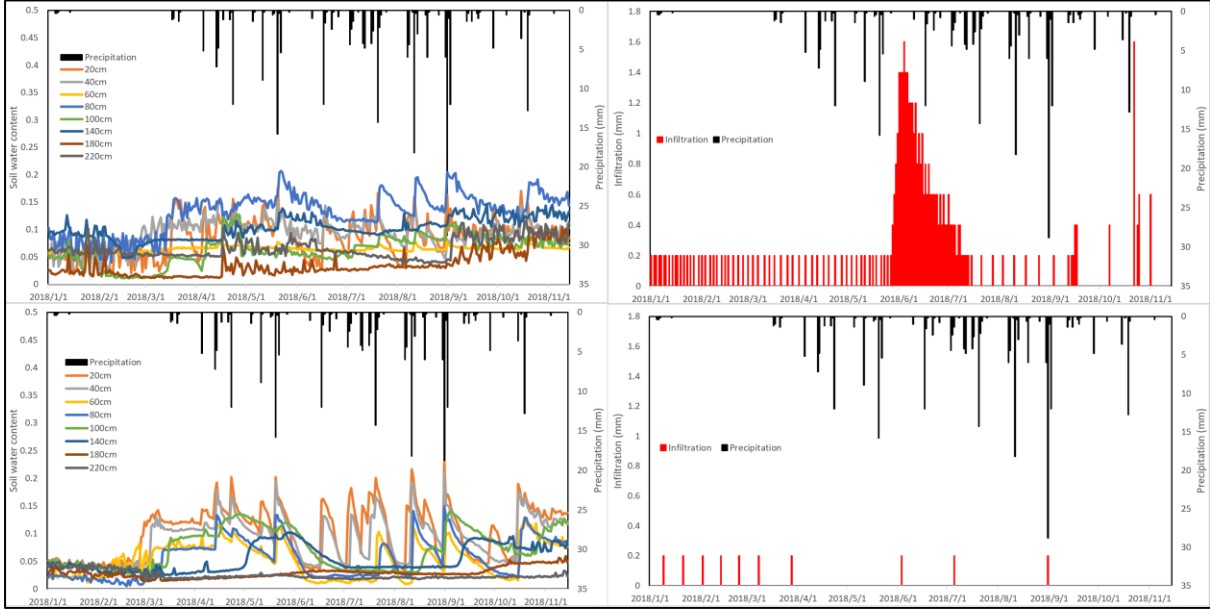

Figure 7 Effects of precipitation on soil moisture (A) and DSR (B) in bare sandy land plot; Effects of rainfall on
soil moisture (C) and DSR (D) on ASK plot, 2018.

### 3.5    Research on rain-feed ASK land water distribution

There are many methods of measure surface layer evapotranspiration, but all have

poor precise, because there are many factors that affect surface layer evapotranspiration

and one cannot consider all impact factors, these factors including vegetation coverage,

environmental and temperature factors. This study treats shallow soil as a whole layer

and measures the amount of surface rainfall recharge, soil water storage and DSR

directly. Based on the directly measured DSR and precipitation, the soil moisture

storage change can be calculated using equation (1). During the five-month intermittent

frozen period, soil moisture sensors provide less reliable soil water content

measurements as the soil moisture sensors are designed to detect liquid water instead

of solid ice. Therefore, this research uses the unfrozen time period from April 1[th] to

November 30[th] to investigate the water distribution. The average soil water contents in

the first week of April and the first week of November are used as the initial and final

values of annual soil water storage, to calculate the change of soil water storage

annually. Based on measured precipitation, DSR and soil water storage, and the water

balance equation (1), the evapotranspiration can be accurately calculated and the results

are shown in Table 4.

In 2016, the soil moisture reserve in the 220 cm soil layer of bare sand increased by

47.15 mm, and the annual evaporation was 134.04 mm, while the soil water storage of

Artamisia plot increased by 31.95 mm, and the evapotranspiration was 342.25 mm. In

2017, the soil water storage of bare sandy plot increased by 13.77 mm, and the annual

evaporation was 232.03 mm, while the soil water storage of Artamisia plot was reduced

by 83.7 mm, and the evapotranspiration was 365.9 mm. In 2018, the soil water storage

of bare sandy plot increased by 72.14 mm, and the annual evaporation was 121.46 mm,

while the soil water storage of Artamisia plot increased by 2 mm, and the

evapotranspiration was 202.63 mm. One should be noted that the change in soil water

storage only represents the distribution of soil moisture from April to November, rather

than the net increase of the whole year, because the water in the soil will continue to

infiltrate to deep soil layer when the surface soil layer is frozen. As shown in Figure 3,

there is no significant precipitation from January to June 2017, but deep infiltration has

been occurring. Comparing the data from 2016 to 2018 in Table 4, it can be found that

when there is sufficient precipitation, for example, in 2016, soil water storage increases

and evapotranspiration increases as well. When the precipitation is low, the soil water

storage decreases and the evapotranspiration decreases as well. The results show that

after vegetation reconstruction in this area, the amount of DSR is significantly reduced,

which may threaten the safety of groundwater recharge; The precipitation water

resource is concentrated in the shallow soil layer, vegetation gets sufficient moisture,

then evaporation increases, and the regional microclimate environment will be

improved. Evapotranspiration of plants in drought years is significantly reduced, which

shows that vegetation will adapt to the environment by increasing or decreasing water

consumption according to the amount of precipitation.

Table 4. Annual water distribution of ASK land and bare sand field

| Year | Field type | Precipitation (mm) | DSR (mm) | Change of the SWS (mm) | Evapotransp iration (mm) |
|------|-----------|-------------------|----------|------------------------|--------------------------|
| 2016 | Bare sand plot | 464.8 | 283.6 | 47.15 | 134.05 |
|      | ASK plot | 464.8 | 90.6 | 31.95 | 342.25 |
| 2017 | Bare sand plot | 313.4 | 67.6 | 13.77 | 232.03 |
|      | ASK plot | 313.4 | 31.2 | -83.7 | 365.9 |
| 2018 | Bare sand plot | 245.2 | 51.6 | 72.14 | 121.46 |
|      | ASK plot | 245.2 | 2 | 40.57 | 202.63 |

*Note: SWS stands for soil water storage.

## 3.6    Influence of vegetation coverage on infiltration rate

In many aspects one can find the influence of vegetation on infiltration, the

interception of precipitation by the aboveground part of vegetation, the interception and

absorption of surface soil layer moisture by vegetation, the absorption and utilization

of soil water by vegetation roots, the root system occupying soil voids to reduce

infiltration speed, and the conduction effect of the catheter formed by death root on the

infiltration ability. In this study, we consider the above-ground and underground parts

of vegetation as a whole system, and compare the bare sand plot and ASK plot on the

infiltration speed. During the observation period, the Precipitation-DSR interaction

occurred alternatively. In order to show the characteristics of the two types of

infiltration, we selected a typical infiltration process, and the result is shown in Figure

8. A precipitation of 90.2 mm/d was generated at 23:00 on July 7, 2016, and a DSR

event was observed at 21:00 on July 9 on the bare sand plot. From the surface soil layer

to 220 cm depth soil layer, the infiltrate process took 46 hours. The DSR of the ASK plot was observed at 8:00 on July 12, and the infiltrate process from the surface layer to the depth of 220 cm soil layer took 107 hours. The infiltration rate of ASK plot is 2.33 times of bare sandy land. One can see that vegetation cover significantly affects the infiltration rate. However, under natural conditions, multiple precipitation processes occur in a short period, so it is difficult to distinguish the DSR event caused by a certain precipitation of different land coverage types under sufficient precipitation.

The results show that the characteristics of precipitation-induced DSR in the sandy land plot and the ASK plot are different. The two precipitation events leave marks on the bare sandy plot, leading to two spikes of DSR. In contrast, such spikes do not appear in the ASK plot, because water is utilized by the root system mostly and only a very small portion of the precipitation-induced infiltration can reach as far as 220 cm to be detected by the lysimeter. ASK not only delays the infiltration rate but also reduces the total amount of DSR.

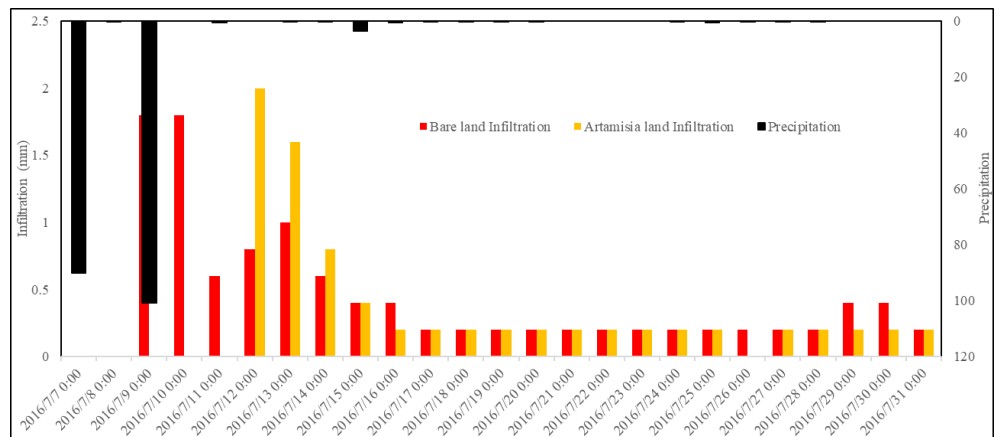

Figure 8. DSR response to precipitation on bare sandy plot and ASK plot

## 4. Discussion

### 4.1 Insights gained from this study

Monitoring and quantifying precipitation induced shallow groundwater recharge process is a long-lasting challenges in the hydrological communities, and it is especially difficult to do so in arid and semi-arid regions because of the spatiotemporally highly variable precipitation and complex soil moisture dynamics during the infiltration process (Newman et al., 2006). Studies of the interrelationship of precipitation and

shallow groundwater is very important to local vegetation reconstruction, with or without anthropogenic mitigation (Ramier et al., 2009;Scibek et al., 2007). The difficulty of attempting to establish a relationship between precipitation and groundwater recharge is mainly reflected in the following aspects. Firstly, there are fewer instruments for direct long-term, large-scale measurements (Krishnaswamy et al.,

2013). The commonly used methods such as double ring filter method, lysimeter, rainfall simulation method, water flux method and stable isotope based tracking methods all have certain specific restrictions (Sprenger et al., 2015;Groh et al., 2018). For instance, the heterogeneous nature of soil and point observations made with most above mentioned methods will make it difficult to conduct a basin scale analysis

(Mousavi and Shourian, 2010). Secondly, ecological elements (such as ASK root systems in this study) are always changing, thus any monitoring methods that cannot continuously accommodate the ecological elements will miss a significant piece of the machinery of understanding the precipitation-recharge relationship. Our research here is an attempt to utilize a low-cost, field-based lysimeter method to monitor DSR for

four years in Mu Us sandy land, a task that measure DSR without disturbing the horizontal flow of soil moisture has never reported before.

       The improvement of soil texture takes hundreds of years. It also takes a long time to improve soil texture with the participation of vegetation roots. In the Mu Us Sandy Land, due to the participation of the sand-fixing vegetation, a large number of dust

particles are fixed by the sand-fixing vegetation. When rainfall happens, water

infiltrates into the deep soil with these dust. Over the forty years, the texture of the soil

in this plot has undergone significant changes. From this one can infer that the speed of

soil improvement in this area far exceeds our expectations.

     In semi-arid areas, Mu Us sandy land as an example, the main limiting factor for

trees is available water resources (Gao et al., 2014;Skarpe, 1991). Therefore, the key to

understand the vegetation ecosystem in semi-arid areas is to study the supply of water

resources (Cheng et al., 2018a;Cheng et al., 2017a). The ASK has been in existence in

the study area for more than 40 years, so the purpose of this study is to find out whether

there is sufficient water resource available in the region to support vegetation ecosystem,

through the measurement of DSR. The "sustainable" growth of plants in this study

means that water resource from precipitation can meet the growth needs of ASK, and

can still have an excess amount of water to replenish deep soil layer. In this study, the

soil moisture distribution has been studied by using the newly designed lysimeter to

measure whether the soil layer below the root layer could produce DSR or not.

In the dry years, the differences in soil water storage and DSR between the two

plots are significant, taking 2018 as an example. At the beginning of the experiment,

the soil moisture storage in the ASK plot is 126.16 mm, and the soil moisture storage

of bare sandy land is 147.22 mm. At the end of the experiment, the soil moisture storage

in the ASK plot is 166.72 mm, which is 40.56 mm less than that at the beginning of the

experiment. The soil moisture storage of the bare sandy plot at the end of the experiment

is 219.37 mm, which is 72.15 mm more than its counterpart at the beginning of the

experiment. There is no significant difference in soil water storage, but the DSR

difference is obvious. The DSR of bare sand is 51.6 mm, and that of ASK plot is only

2 mm. Although the DSR is significantly reduced, even in the dry years, there is still a

small amount of DSR, indicating that the selection of ASK as sand-fixing vegetation in

this area is a suitable plant species. Another interesting point to note is that ASK is

capable of adjusting their own growth conditions based on the available moisture

recharge, and a larger moisture recharge will result in a faster growth rate of such plants.

When the rainfall is insufficient, the evapotranspiration of ASK is reduced from 342.25

mm in 2016 to 202.63 mm in 2018.

As surface soil is frozen and ASK enters dormancy during winter in the research

site, snow can only accumulate on the surface and cannot recharge soil moisture.

However, moisture in deep soil continues to infiltrate downwards because of the driving

force of gravity. This is particularly true in bare sandy land as a large amount of soil

moisture has been accumulated at the start of the frozen period. A portion of those

accumulated soil moisture will slowly infiltrate downwards and recharge groundwater

reservoir. Because the amount of snowfall in winter is difficult to calculate, the amount

of frozen water accumulated in winter cannot be obtained.

Traditional lysimeter have impervious containers for loading plants. The air in the

container will cause increased DSR when it expands and contracts. Some researchers

believe that it is condensed water generated by soil water vapor. The new designed

lysimeter has improved this design and no longer has the interference of condensate for

there is no container. But it should also be mentioned that condensed water does exist

in sandy soil layer and it is not easy to measure. We also need to mention that although

vegetation in arid areas grows slowly, the amount of water that becomes plant tissue

during the growth is relatively small, and most of the water is consumed by

evapotranspiration. The tissue water consumed by vegetation is not calculated

separately in this research but is counted in evapotranspiration.

**4.2 Limitations and future works**

We need to point out that airflow and vapor transport has not been considered in this experimental investigation and it should be incorporated in future studies because of two considerations.

      Firstly, it has been reported that airflow may play an important role for mass and energy transport in arid and semi-arid areas (e.g., Scanlon, 1994; Scanlon and Milly, 610   1994; Zeng et al., 2009a, 2009b; Zeng et al., 2011a, 2011b; Yu et al., 2018, Yu et al., 2020), this is particulary true when discussing evaporation process. For instance, Zeng et al. (2011a) has established a one-dimensional (vertical) two-phase heat and mass flow model to explain the field measurements of soil moisture content and temperature in the Badain Jaran Desert of China for both low- and high-permeability soils. They 615   reported that the evaporation was underestimated when the airflow was neglected and such underestimations were more evident in the low-permeability soil (6.4%) as in the high-permeability soil (8.85%). Zeng et al. (2011a) concluded that such underestimations of evaporation were mainly caused by underetimation of isothermal hydraulic conductivity by neglecting airflow. Mohanty and Yang (2013) agreed with 620   Zeng et al. (2011a) that the underestimation of evaporation was caused by underetimation of isothermal hydraulic conductivity, but they diagreed with Zeng et al. (2011a) that negligience of airflow was responsible for the underestimation of isothermal hydraulic conductivity. The critical comment made by Mohanty and Yang (2013) was disputed by Zeng and Su (2013) who upheld the conclusions of Zeng et al. 625   (2011a), but at the same time Zeng and Su (2013) recognized that some other mechanisms such as adsoption of component of the soil water retention (which has been pointed out by Mohanty and Yang (2013)) can be important and should be included in addition to diffusion, advection, and dispersion when discussing the balance equations

of water (liquid and vapor), dry air, and heat. In summary, there is a general consensus that airflow is relevant when discussing the mass and energy transport in the unsaturated zone, particularly near the land-atmospheric surface. However, it is still not fully understood to what degree the airflow has contributed to the land-atmosphere interaction.

Secondly, this study mostly concerns liquid water movement below the shallow soil zones (like 2.2 m below ground surface) with a water table as deep as 7 meters in an semi-arid region. How important is vapor transport in the study site is an open question that should be answered when new experiments are conducted in the future. It is speculated that even for ground watertable as deep as hundres of meters, continuous upward vapor transport either driven by thermal gradient, soil matrix potential, or diffusion and dispersion processes may still exist and can be important source of water in desert area like the site of this study (Scanlon, 1994; Scanlon and Milly, 1994). In the past decades, relevant studies in Badain Jaran Desert of China have indicated that vapor transport has played an important role in regulating infiltration and land surface evaporation (Zeng et al., 2009a, 2009b; Zeng et al., 2011a, 2011b; Zeng and Su, 2013). In addition, the vapor transport is also important for freeze-thaw cycles in the Badain Jaran Desert (Yu et al., 2018, Yu et al., 2020). In summary, further research is needed to quantify the importance of airflow and vapor transport and source of condensation at the study site.

## 5. Conclusions

This research uses a newly designed lysimeter to monitor shallow soil layer infiltration, and results show that in order to absorb more precipitation moisture, ASK develops a horizontal root system and retains more water in the shallow soil layer. ASK

has shown to be effective in fixing the mobile sand and increasing the proportion of fine particles in the sandy land. ASK changes its own evapotranspiration mount to adapt to the annual precipitation changes. Under the existing precipitation conditions, the ASK community can develop healthily, as a small amount of precipitation can recharge the groundwater, even in dry year. This indicates that precipitation in the area is sufficient to meet the needs of vegetation water consumption. However, with the unforeseeable global warming and abnormal precipitation events, semi-arid region may become drier and the ASK community may be seriously affected. Therefore, continuously monitoring the key controlling factors associated with the ecological system in the semi-arid region is needed.

The following conclusions can be drawn from this research:

1)  In Mu Us sandy land, the ASK root system develops horizontally to absorb more precipitation-induced infiltration. The root system reaches 120 cm depth and mainly concentrates within the upper 40 cm deep soil layer.

2)  After 40 years of vegetation reconstruction, the soil particle size distribution has been significantly changed; soil texture improvement in semi-arid sandy land far exceeds expectations. Specifically, the sandy soil mainly consisting of medium sand (50.53%) grows into a sandy soil mainly consisting of fine sand (60.53%). Vegetation is particularly important in semi-arid areas since it directly changes the composition of soil.

3)  The yearly DSR in the ASK sand-fixing experimental plot is from 2 mm to 90.6 mm. In contrast, the yearly DSR in the bare sand plot is from 51.6 mm to 283.6 mm. This shows that the rainfed vegetation has reduced DSR substantially but there is still a small amount of recharge left to replenish the deep soil moisture, implying that the current ASK community is still hydrologically self-

sustainable because it does not consume all the water moisture replenished by precipitation and the DSR has not been reduced to zero.


4) With the coverage of ASK at this stage, the sand-fixing forest in the Mu Us Sandy Land can still achieve connectivity between atmospheric water, surface water and ground water. During the observation period, the infiltration of bare sandy land can reach 25.8 times of ASK land, and during a single precipitation process, the infiltration rate of ASK plot can reach 2.33 times of bare sandy land.


The airflow and vapor transport have not been considered in this study and should be incorporated in future experimental and theoretical investigations at the site.

**Acknowledgements**

This research was supported by the National Key R&D Programe of China(2016YFE0202900). Funding of basic scientific research operations of the

Chinese Academy of Forestry (CAFYBB2020MB007). Open Project Program of Ministry of Education Key Laboratory of Ecology and Resources Use of the Mongolian Plateau(KF2020004). Major Program of the National Natural Science Foundation of China, National Science and Technology Major Project and Ministry of Science and Technology of China (No. 2017ZX07101004, No. 2018YFC0507100, No. 2019ZD003,

NO. 31971726), the National Non-profit Institute Research Grant of Chinese Academy of Forestry (grant number CAFYBB2018ZA004) and the research grants from the Fundamental Research Funds for the Central Universities (BLX201814, No. 2015ZCQ-SB-01). We gratefully acknowledge the Beijing Municipal Education Commission for their financial support through Innovative Transdisciplinary Program "Ecological

Restoration Engineering". Thanks to the experimental site provided by Inner Mongolia Dengkou Desert Ecosystem National Observation Research Station, the Experimental

Center of Desert Forestry, Chinese Academy Forestry. The authors thank two anonymous reviewers for their constructive comments which help improve the quality of the manuscript.

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
