# Peer review of "New measures of deep soil water recharge during vegetation restoration process in semi-arid regions of northern China"

_Hydrology and Earth System Sciences, 2020_

## Referee Comment (RC1) · Anonymous Referee #1 · 27 Jul 2020

Via measuring deep soil water recharge during vegetation restoration process in Mu Us Sandy land, and relevant root system survey, soil texture analysis, and soil moisture monitoring, this study tries to understand the difference of infiltration processes between re-vegetated and bare sandy land. Although the manuscript is generally well structured and written, this reviewer found only little contribution of this study to the community. Particularly, the plant type and relevant root distribution as controlled by precipitation infiltration depth (&rate) and capillary rise height (&rate) from groundwater is well known and has been summarized by Yin Fan (2017, PNAS). Perhaps, the most novel part of this study is the measurement of DSR using a new instrument. On the other hand, the explanation and discussion on this instrument is very limited.

Some major concerns are as follow: 1) The author claims that the DSR measurement performed in this study, being a task never reported before. On the other hand, the explanation on the DSR measurement and relevant principles/details were not presented. For example, it is understood a lysimeter installed at the depth of 3.2 meter, to enable the measurement of DSR as being below the deepest root c.a. 100cm. However, the length and width of this lysimeter is only 0.3m*0.3m, while the root distribution can reach a 200cm diameter horizontally. This makes this reviewer questioning the reliability of DSR measurements. Also within the deep soil, the relative humidity is rather hight (e.g., 99.9%), which will lead to vapor condensation on lysimeter device, how this vapor condensation effect is removed also needs some explanations. 2) The author claims that the direct measurement of ET is not reliable, but the current approach deployed to measure DSR combined with water balance equation will give accurate estimation of ET. This is a very strong statement while way beyond the reality. If one looks back the point one about the reliability of DSR measurement in this study. 3) The way the author investigate the soil texture change is too much data-limited (e.g., only one plot, and the averaged mixed information was used), which renders the reliability of relevant analysis. 4) There are no any numerical analysis/experiment to investigate/validate relevant hypothesis, which also jeopardized the credibility of this study.

Please see some minor comments attached.

Please also note the supplement to this comment:
https://hess.copernicus.org/preprints/hess-2020-200/hess-2020-200-RC1-supplement.pdf
* * *
[Figure]

**Supplement:**

[revised manuscript text omitted]

---

## Referee Comment (RC2) · Anonymous Referee #2 · 1 Aug 2020

The Three North Shelterbelts is a huge afforestation program launched in China in the last century, which has made great contributions to regional sand fixation, dust-storm prevention and ecological environment improvement. However, it caused great concerns about hydrological cycle and ecological environment evolution. In this manuscript, the characteristics of water cycle were obtained through the comparative analysis of the observation data of the key parameters in the process of hydrological cycle in the Artamisia sphaerocephala Krasch sand-fixing land in the Three North Shelterbelt area. The basis of this research is the formula (1) in L160, as the evapotranspiration, the manuscript stated "Evapotranspiration is calculated through a water balance equation when precipitation and soil moisture data are collected", how is

it calculated? whether the calculation is accurate. Is DSR=P+Cm*d–E? ±âŰşW a soil moisture storage change (should be variation here), what is the physical basis or meaning? W is soil moisture storage, what is thisïij§ what form it is storedïij§ Since the horizontal growth of shallow roots is concluded, is it reasonable for the Lysmeter not to consider the horizontal soil water transformation? In addition, how many samples of Artamisia sphaerocephala Krasch sand root excavations? How representative is it? Before accepting for publication, all these questions need to be implemented by the authors.

Specific questions

L16-all other lines: Why do not use the Eddy Covariance System to measure the near surface evapotranspiration? At least the calculated values should be validated by this observation.

L72-other lines: The full text should use the passive voice, because characteristics of Artamisia sphaerocephala Krasch are not developed by itself, but formed by environmental forcing.

L133: Since the Artamisia sphaerocephala Krasch developed horizontal root, is it too small to excavate a length and width both of 0.3m soil column or design such size of a lysmeter?

L163: How the DSR is measured or estimated should be specified here.

L206: It makes sense to analyze the changes of soil organic matter.

L240: How to physically define the thawing recharge period, germination consumption period, rain season recharge period and plant dormancy period and the frozen soil periodïij§

L245: Figure 3 should show the soil temperature curve to identify whether it is soil thawing period.

L504: What is "sufficient precision" means? I'm sure that 2.33 will be changed with a different "sufficient precision".

---

## Author Response (AR1)

We thank anonymous reviewer for their constructive comments. The manuscript has been significantly improved by addressing the comments. The following are our point-to-point responses to their comments.

**Responses to the Comments from Reviewer #1**

*1) The author claims that the DSR measurement performed in this study, being a task never reported before. On the other hand, the explanation on the DSR measurement and relevant principles/details were not presented. For example, it is understood a lysimeter installed at the depth of 3.2 meter, to enable the measurement of DSR as being below the deepest root c.a. 100cm. However, the length and width of this lysimeter is only 0.3m\*0.3m, while the root distribution can reach a 200cm diameter horizontally. This makes this reviewer questioning the reliability of DSR measurements. Also within the deep soil, the relative humidity is rather hight(e.g., 99.9%), which will lead to vapor condensation on lysimeter device, how this vapor condensation effect is removed also needs some explanations.*

**Response:** Thank you for the comment. The explanation of this new type of Lysimeter has been published in HESS in details before (Cheng et al, 2017), so that is why it is only briefly explained in this article. The conventional Lysimeter uses an impermeable container (constructed all the way from ground surface downward) to wrap the soil column, blocking the horizontal flow of the soil layer, thus there is inevitable water potential difference existing inside and outside the container of the Lysimeter. It is notable that horizontal soil moisture flow in the active root zones in arid and semi-arid regions could be significant as the roots prefer to grow horizontally to intercept the maximum infiltrated water. In arid and semi-arid regions, the roots usually do not grow vertically to great depths because the regional groundwater table is so deep that it is almost impossible for roots to tap groundwater. Below the active root zones, horizontal movement of water moisture will be substantially reduced and vertical movement of water moisture starts to prevail.

The new Lysimeter has an upper water balance part and a lower measurement part which can directly measure the water flux. Specifically, the flux infiltrating into the balance part at the depth of the measurement face should equal the flux exiting the balance part and entering the measurement part. There is no need to build an impervious container to wrap the vegetation tested for the new Lysimeter above the measurement face. The 0.3 m*0.3 m is the planar view size of the new Lysimeter, and the height of the Lysimeter is 1 meters. Because the measurement face of the Lysimeter is at a depth of 2.2 meters, the installation of the instrument requires downward excavation to a depth of 3.2 meters. After this step, lateral excavation will be conducted (under a 2.2 m undisturbed soil) to generate a cave with the size of 0.3 m of width, 0.3

m of length, and 1 m of height to host the Lysimeter. After the installation of the Lysimeter, the excavation will be backfilled using the native soil. The experimental site was flat sandy land before ASK was planted for sand control 40 years ago. After 40 years of development, the region is dominated by ASK, scattered Rhamnus parvifolia, Chenopodium glaucum, Setaria viridis and the field average vegetation coverage has reached 80%. In order to investigate the distribution of the roots of ASK, five ASK plants with the same growth and age were excavated on the adjacent plots of the experimental site to conduct the analysis of roots layer by layer up to 1.2 m below ground surface. The average root of these five ASK plants at a particular depth is regarded as the representative root of ASK plants in this area at that depth.

The condensate of the soil layer in the semi-arid area is indeed an important source, but the groundwater level in this area is about 7 meters deep, so the groundwater replenishment on soil moisture for shallow soil layers less than 2.2 m deep is essentially negligible. Therefore, the source of condensate at a depth of 2.2 meters comes solely from precipitation, and we can still use the water balance principle to calculate the distribution of precipitation-induced infiltration in each soil layer.

*2) The author claims that the direct measurement of ET is not reliable, but the current*

*approach deployed to measure DSR combined with water balance equation will give accurate*

*estimation of ET. This is a very strong statement while way beyond the reality. If one looks*

*back the point one about the reliability of DSR measurement in this study.*

**Response:** Implemented. According to the literature review conducted by the authors, the methods of directly measuring ET include Lysimeter, eddy correlation method, Bowen ratio method, Large aperture scintillation method, etc. Taking the most advanced eddy correlation method as an example, the measurement error may be 20% or higher and the required monitoring conditions are quite demanding. Furthermore, it is difficult to avoid the influence of human factors on the experimental results. This study provides an inexpensive measurement method that directly measures the water flux at the lower interface of the target layer (the deep soil recharge or DSR), and combine DSR with  a few factors that can be accurately measured in real applications (such as precipitation and soil water storage) to calculate ET from the law of conservation. The deficit of this method is that it measures the ET at the point where the

Lysimeter is located, thus it may not be the representative ET value over a large scale. Upscaling of the point ET value to large-scale ET value is an important issue and should not be overlooked in the future.

*3) The way the author investigate the soil texture change is too much data-limited (e.g., only*

*one plot, and the averaged mixed information was used), which renders the reliability of*

*relevant analysis.*

**Response:** First of all, we need to point out that this study does not consider the exact spatial distribution of soil size distribution that is often important for conducting a precise unsaturated flow simulation (which is not the focus of this investigation). Instead, the purpose of measuring the soil particle sizes is to estimate the capillary rise based on the "average" grain sizes of sand.

The capillary rise height is an important parameter for designing the new Lysimeter (as the height of the balance part of the Lysimeter should be greater than the typical capillary rise height at the site). Secondly, the artificially planted ASK forest land of the experimental plot is relatively homogenous. We selected five pieces of ASK for excavation and collected soil samples layer by layer over the 220cm thick soil profile. To study the changes of soil particle sizes over depth, we mixed the soil of five samples at individual layers and analyzed it to minimize errors caused by one plot sample.

*4) There are no any numerical analysis/experiment to investigate/validate relevant hypothesis,*

*which also jeopardized the credibility of this study.*

**Response:** At present, there are many models to study the process of soil infiltration, but there are relatively few measured data for arid and semi-arid sandy lands. The purpose of this study is to use the newly designed Lysimeter to measure the water balance information of precipitation in this special type of land, and to evaluate the deep soil water information of the area through the measurement of DSR. The probes used here are commonly adopted by many other soil scientists and are reliable. It is our intention that such acquired datasets can be eventually utilized in sophisticated numerical modeling of unsaturated zone water dynamics in the future.

**Response to other comments in the article**

*1   line 60, line 75, Citation error*

**Response:** Implemented.

*2   line 95, The bottom right picture has glare effect, cannot give readers nice impression on*
*    the experimental site. Also, it would be nice to have a UAV image of the study site.*

**Response:** Implemented. We do not have a drone, but we chose a clear picture of the plot
instead.

*3   line 105-107, This is very confusing. Are you saying you have a groundwater table at*
*    180cm? and 60cm capillary rise is from that 180cm GW table? But you said the region has*
*    GW table of 5.3 - 6.8m. This deserves careful consideration and clarification. Studies have*
*    shown that the vapor can be transferred to surface from more than 100m deep zone of the*
*    sand dune.*

**Response:** The depth of the root layer of Artemisia Ordosica in this area is 120 cm (meaning that water within the 120 cm depth may be transpired through root). Furthermore, the capillary water rise height of the sandy soil in this area is 60 cm (meaning that water may be moved upward a maximum height of 60 cm by the capillary force). Therefore, the maximum uplifting of water through the transpiration of root and capillary rise is the summation of 120 cm and 60 cm, which is 180 cm. In another word, for any water within the 180 cm depth, there is a possibility that it can return to atmosphere (The amount of vapor transmission is small, so it is not considered here); for any water below 180 cm depth, it is impossible for it to return to atmosphere (thorough evapotranspiration) and it can only keep going down to recharge the deep soil, assuming that vertical downward movement of soil moisture below the 180 cm depth is dominating and any horizontal soil moisture movement below the 180 cm depth is secondary and negligible. In this study, considering the possible (but unlikely) small variation of capillary rise because of the (minor) soil particle sizes variation in the space, we further extended the depth of measurement

40cm down to 220cm below ground surface. By doing so, the possibility for water at the depth of

220cm t5.3-6.8 m o return to the atmosphere through evapotranspiration is essentially zero.

Above analysis will not be affected by the regional groundwater table which is sufficiently deep.

*4    line 140, what is the wet bulk density you are referring to? before and after backfill? Are*

*they close to each other? If not , how will this affect your results?*

**Response:** This paper does not concern the accurate determination of bulk density. The description here is that water is used to wet the soil before excavating the soil profile and installing the Lysimeter to make sure that the sand is relatively compact and will not collapse during the excavation process. When installing the Lysimeter, excavation is conducted outside of the targeted area first to a designated depth and then conducted sideway horizontally, and the

Lysimeter is installed under the undamaged soil layer above the Lysimeter. After the Lysimeter is installed, water is also used to wet the soil profile to make sure that the sand is relatively compact. The collected data can be used after the sand has settled naturally for one year to ensure data quality.

*5    line150, The dimension/scale is not indicated in the figure. Please do.*

**Response:** Implemented.

*6    line 153, well, this deserves more consideration. In Land Surface Model, if you considered*

*most of the physical processes (e.g,. intercepted water by leaf, interception-induced*

*evaporation, through-fall, soil hydrothermal properties etc.) you should be able to estimate*

*infiltration rate and runoff. This is very basic though.*

**Response:** If all factors are taken into consideration, it is indeed possible to predict whether there will be runoff on the surface. However, the reality is that the environmental factors are complex, with large changes in vegetation coverage, precipitation, and soil moisture. This is indeed an important issue to address but further data collection works are needed to make it possible. The research here is based on direct field observations, and the terrain in this area is relatively flat and there is no surface runoff. The focus of this research is to explore the water balance process of precipitation water in unknown environments using a simple and straightforward water balance approach based on direct observation of DSR and other factors.

*7   line 254, Do you have measurement of transpiration? I guess you mean root water uptake*

*here? Do you mean all the soil moisture decrease can be attributed to RWUP here?*

**Response:** This study did not separate transpiration from evapotranspiration, and used evapotranspiration to classify plant evaporation and soil transpiration as one category. The reduction in soil water content of each layer here should be the result of root water uptake and soil evaporation. From April 25th to June27th, there were 31 observed precipitation events in total. The maximum precipitation was 18.8 mm, and the minimum precipitation was 0.2 mm.

These precipitation events did not change the decreasing trend of soil moisture. This study is incapable of tell whether soil water consumption during this period is transpiration or plant water consumption (root water uptake). However, the soil moisture drops sharply during the germination period. The transpiration intensity of this period is not the largest in an annual basis.

The summer soil transpiration intensity is greater but we still observe that precipitation infiltrates to recharge deep soil. Based on this, we speculate that vegetation consumes a great deal of soil water during the germination period, and the specific amount of water consumed needs further and more detailed experimental observations.

*8   line 280, Do you have long climatology to be compared with? In order to determine wet,*

*dry and normal.*

**Response:** As shown in line 90 in the text, the precipitation observation data from 1960 to 2010

in this area show that the multiple-year average precipitation is 358.2 mm. When the annual precipitation at a particular year is higher than 358.2 mm, it is considered a wet year; if the annual precipitation at a particular year is lower than 358.2 mm, it is considered a dry year.

*9   line 287, This is not correct and not corresponding to the content of this section. Please*

*rephrase. It is more "characteristics of DSR ... ..."*

**Response:** Implemented.

*10  line 355, It is more water balance analysis than water distribution.*

**Response:** Implemented.

*11   line 460, You mean even only 2mm DSR is enough to sustain the local water demand other*

*than sustaining the local ecosystem itself. It is suggested to explain a bit more details in*

*terms of 'sustainability'. Do you consider plant only here or also local water resources for*

*other use?*

**Response:** The DSR that can enter the 200 cm depth soil layer is the remaining water after the consumption by vegetation. If the DSR is greater than zero, it means that precipitation not only can meet the needs of vegetation growth, but also has excess water that can infiltrate into the deep soil layer. When this is the case, we regard the system as sustainable. Otherwise, if the DSR

is reduced to zero, it means that the precipitation is not sufficient for satisfying the consumption of the vegetation, thus is unsustainable.

**We thank anonymous reviewer for their constructive comments. The manuscript has been**

**significantly improved by addressing the comments.**

**Responses to the Comments from Reviewer #2**

*The Three North Shelterbelts is a huge afforestation program launched in China in the last*

*century, which has made great contributions to regional sand fixation, dust-storm prevention*

*and ecological environment improvement. However, it caused great concerns about*

*hydrological cycle and ecological environment evolution. In this manuscript, the*

*characteristics of water cycle were obtained through the comparative analysis of the*

*observation data of the key parameters in the process of hydrological cycle in the Artamisia*

*sphaerocephala Krasch sand-fixing land in the Three North Shelterbelt area. The basis of this*

*research is the formula (1) in L160, as the evapotranspiration, the manuscript stated*

*"Evapotranspiration is calculated through a water balance equation when precipitation and*

*soil moisture data are collected", how is it calculated? whether the calculation is accurate. Is*

$P + C_m*d - DSR - E = \pm \triangle W$ *a soil moisture storage change (should be variation here), what*

*is the physical basis or meaning? W is soil moisture storage, what is this? What form it is*

*stored? Since the horizontal growth of shallow roots is concluded, is it reasonable for the*

*Lysmeter not to consider the horizontal soil water transformation? In addition, how many*

*samples of Artamisia sphaerocephala Krasch sand root excavations? How representative is it?*

*Before accepting for publication, all these questions need to be implemented by the authors.*

The purpose of this research is to use a newly designed Lysimeter to directly measure the deep soil recharge (DSR) of the ASK plot at the depth of 2m without damaging the in-situ soil layers.

Based on the obtained information of DSR at 2 m depth, the change in soil moisture content from the beginning of the experiment to the end of the experiment and precipitation amount, then evapotranspiration can be calculated by using a water balance equation. This is a new method to obtain DSR information at a targeted depth of soil layer based on direct field observation in arid and semi-arid sandy land when the regional water table is sufficiently deep so will not affect the measurement of DSR.

The advantage of this newly designed instrument is that it can be directly installed at a depth of 2

m depth, and there is no need to wrap a soil column like a conventional Lysimeter to block the horizontal flow of soil. There are no outflow rivers and artificial recharge in this area, and precipitation is the only source of water recharge in this area. Considering that 99% of the water consumed by vegetation is evapotranspiration (ET), the residual water remained in plant structure could be ignored.

According to the principle of water balance, precipitation = ET+ the change of soil water storage within 2m + the amount of DSR. Precipitation is measured by a rain gauge, DSR is directly measured by this new type of Lysimeter, and the soil moisture storage within 2 m is obtained by the soil moisture probes to obtain the soil volumetric water content, which is multiplied by the thickness of the soil layer to yield the soil water storage. Therefore, ET can be computed using above water balance equation.

The experimental plot is flat, and the soil is relatively uniform in the horizontal direction, the coverage of the plot reached 80%, the plot is relatively homogeneous, so we can use experimental observation result at a local scale as a surrogate to represent the entire homogeneous area. However, upscaling of the point ET value to the large-scale ET value should be cautious and not overlooked in the future investigations. The new instrument avoids disturbing the soil layer and directly measured the DSR at the soil interface at a depth of 2 m.

In this experiment, five adjacent ASK samples are selected for excavation when collecting the root samples of ASK, and the mean value of the roots of each layer is used for analysis. The purpose of collecting this parameter is to explore the depth of the roots of the ASK and to determine the buried depth of the new Lysimeter. This study provides a new method for measuring the DSR in arid and semi-arid regions, and based on this information, evapotranspiration could be calculated.

Despite the fact that this research is primarily experimental, such experimental works have not been carried out before to investigate the ecohydrological consequence after planting ASK  in semi-arid regions and it serves as an important experimental framework of testing soil moisture movement dynamics theories in the future.

**The following are our point-to-point responses to their specific comments.**

*1 L16-all other lines: Why do not use the Eddy Covariance System to measure the near*

*surface evapotranspiration? At least the calculated values should be validated by this*

*observation.*

**Response:** This is a very valuable suggestion that can be incorporated in further investigations.

Eddy covariance system is a method of measuring evapotranspiration, and this method is mostly used in large ecological observation stations and is usually quite expensive. Up to present, it has not been implemented in most semi-arid areas of China. In the future, we will conduct a similar experiment near an Eddy covariance system station and compare the data with the Eddy covariance system data.

*L72-other lines: The full text should use the passive voice, because characteristics of*

*Artamisia sphaerocephala Krasch are not developed by itself, but formed by environmental*

*forcing.*

**Response:** Implemented. The text has been revised accordingly.

*L133: Since the Artamisia sphaerocephala Krasch developed horizontal root, is it too small to*

*excavate a length and width both of 0.3m soil column or design such size of a lysmeter?*

**Response:** Thank you for the comment. The conventional Lysimeter uses an impermeable container (constructed all the way from ground surface downward) to wrap the soil column, blocking the horizontal flow of the soil layer in the root zones(see supply figure A, the conventional Lysimeter; figure B, the new Lysimeter). It is notable that horizontal soil moisture flow in the active root zones in arid and semi-arid regions could be significant as the roots prefer to grow horizontally to intercept the maximum infiltrated water. In arid and semi-arid regions, the roots usually do not grow vertically to great depths because the regional groundwater table is so deep that it is almost impossible for roots to tap groundwater. Below the active root zones, horizontal movement of water moisture will be substantially reduced and vertical movement of water moisture starts to prevail. Meanwhile, if a conventional Lysimeter is used, the vegetation needs to be transplanted into the container, so the soil structure and the vegetation root system will be disturbed. The new Lysimeter of this study is designed to be a small-sized instrument installed at any targeted depth of soil layer below the active root zones in arid and semi-arid regions, without blocking possible horizontal water moisture movement in the active root zones.

The plot selected for the experiment is the artificially restored ASK sand-fixing land. The terrain is widely distributed in the Mu Us sandy land and is relatively flat. ASK is the main vegetation species and the soil types is sandy soil. Under these rather "homogeneous" conditions, the experimental result of selected plot may be representative of the ASK sand-fixing land region.

However, we do want to point out that one should be cautious for conducting any upscaling of local value of evapotranspiration to large-scale evapotranspiration value.

The groundwater is deeply buried in the selected plot (around 5.3-6.8 m deep) so the roots of

ASK (which is usually less than 1.2 m deep) cannot tap groundwater for water supply. Therefore, precipitation becomes the only water source supply for ASK to grow. In order to maximize the water taking capability, the ASK root system develops horizontally, making the root layer of the plot relatively evenly distributed in a planar view. The other difference from the traditional

Lysimeter is that the measuring face of the new Lysimeter is not limited to the ground surface. In fact, this newly designed Lysimeter can be embedded in any depths below the active root zones.

*L163: How the DSR is measured or estimated should be specified here*

**Response:** The amount of DSR can be obtained directly through the newly designed Lysimeter,
please see (Cheng et al., 2017), also described in this paper, please see line 155-172.

*L206: It makes sense to analyze the changes of soil organic matter.*

**Response:** Thank you for the comment. Artificially planted ASK has significantly changed the
composition of the soil in the studied area and the distribution of soil organic matter in the top
soil. We have tested the organic matter and obtained relevant data for soil layer with the upper
200 cm soil profile, and we will add this new information and analysis in the revised version of
this paper. The soil organic matter of the ASK plot is higher than that of the bare sandy land plot
at any specific depths within the active root zone. As the depth increases, the soil organic matter
of the ASK plot decreases significantly. The soil organic matter content in the 0-20cm depth soil
layer is the highest, reaching 1.92g/kg, and the soil organic matter at the depth of 200cm depth
soil layer is only 1.5g/kg. The soil organic matter is certainly an important factor, more soil
organic matter distribution information will be studied in details when investigating the
ecohydrological aspect of the plants in semi-arid regions.

The purpose of this research is to use a newly designed Lysimeter to measure the amount of DSR
in order to explore the water balance in arid and semi-arid regions. The soil particle size is
measured in this paper for the sole purpose of estimating the height of capillary rise, which will
subsequently determine the height of the balance part of the new Lysimeter.

*L240: How to physically define the thawing recharge period, germination consumption period,*
*rain season recharge period and plant dormancy period and the frozen soil periodïＪj§*

**Response:** Thank you for the comment. In this study area, the freeze-thaw period refers to the
topsoil (2 meters depth) from the beginning of freezing to the complete melting of the frozen soil
(from October to April). The germination period begins from the end of freeze-thaw period to the
period when branches of ASK are enlarged, and one or two new leaves start to grow (from April
to June). The rainy season refers to a period of relatively concentrated precipitation experienced
after the germination period of ASK plot in this region (from June to October). The analysis of
the soil water replenishment in the freeze-thaw period, the germination period and the rainy season in this area helps illustrate the replenishment characteristics and water sources of soil moisture and DSR at different times. We will add references to show soil temperature information based on other studies which have reported soil temperature monitoring experiments in similar plots like this study.

***Figure 3 should show the soil temperature curve to identify whether it is soil thawing period.***

**Response:** We have designed a double-tube apparatus to measure the depth of the freeze-thaw layer depth. The double-tube apparatus consists of a hollow barrel with a vent hole and a rubber tube inside the barrel. First, the hollow barrel with the vent hole is buried vertically in the experimental plot. The upper level of the barrel is at the ground surface and the length of the barrel is 200 cm. The rubber tube filled with water is placed in the cylinder, and it will be extracted at 8PM daily to record the soil frozen state and the frozen depth, which can be used to interpret the thickness of the frozen soil at that moment. This method can observe the thickness of frozen soil layer, but no soil temperature curve.

***L504: What is "sufficient precision" means? I'm sure that 2.33 will be changed with a***

***different "sufficient precision".***

**Response:** Thank you for the comment, the sufficient precision should be changed to wet year.

The wet year means that the precipitation amount of this year is higher than the multi-year average precipitation amount. In 2016, the annual precipitation amount is greater than the multi- year average precipitation amount, so it is regarded as a wet year. Under the conditions of wet year 
[revised manuscript text omitted]

      2011.

---

## Referee Report (RR1)

1. Line 44-49. The author responds that
   "*The condensate of the soil layer in the semi-arid area is indeed an important source, but the groundwater level in this area is about 7 meters deep, so the groundwater replenishment on soil moisture for shallow soil layers less than 2.2 m deep is essentially negligible. Therefore, the source of condensate at a depth of 2.2 meters comes solely from precipitation, and we can still use the water balance principle to calculate the distribution of precipitation-induced infiltration in each soil layer.*"

   First of all, as I commented, even the groundwater is very deep down to hundreds of meters, you still can find continuous upwards vapor flow (either driven by thermal gradient or soil matric potential gradient)[1,2]. This is essentially important source of water in desert area like your study site. There are also relevant studies implemented over the Badain Jaran Desert, indicating the important role of vapor transport in the sand, and how it affects infiltration and land surface evaporation[3-7]. Furthermore, the vapor transport is also important for freeze-thaw cycles in soil[8,9]. As such, the statement of "the source of condensation at a depth of 2.2 meters comes solely from precipitation" cannot be hold.

   To this reviewer, it is ok you can use simplified equation to derive how much water leave the root zone, and how much infiltrated by precipitation etc.. On the other hand, the total ignorance on the importance of other mechanisms will let readers (incl. this reviewer) to question the scientific rigorousness on this study.

2. Line 108. "the amount of vapor transmission is small, so it is not considered"
   Over moist region, it is ok to neglect this vapor transport mechanism. However, in arid and semi-arid area, the vapor transport is dominant, or at least the same important as the liquid flow in the soil. See my point 1 and relevant literatures.

3. Line 109 "It is impossible for it to return to atmosphere … and it can only keep going down to recharge the deep soil"
   This is a very rough and not necessarily correct statement. As here, the author only assume gravitational potential and ignore the soil water flow driven by soil matric potential gradient.

   Although the author assumed that 60cm is the capillary rise, as such, any water below 180cm is essentially kept in the soil. This assumption is very strong, and lacking rigorous physically-base proof. Furthermore, in this site, you still have a thick unsaturated zone between the bottom of 1.8m to the groundwater table 5-7m. Such assumption of all soil water within this thick unsaturated zone is purely driven by gravity is essentially not correct.

References:

1    Scanlon, B. R. WATER AND HEAT FLUXES IN DESERT SOILS .1. FIELD STUDIES. *Water Resources Research* **30**, 709-719, doi:10.1029/93wr03251 (1994).

2    Scanlon, B. R. & Milly, P. C. D. WATER AND HEAT FLUXES IN DESERT SOILS .2. NUMERICAL SIMULATIONS. *Water Resources Research* **30**, 721-733, doi:10.1029/93wr03252 (1994).

3    Zeng, Y. & Su, Z. Reply to comment by Binayak P. Mohanty and Zhenlei Yang on "A simulation analysis of the advective effect on evaporation using a two-phase heat and mass flow model". *Water Resources Research* **49**, 7836-7840, doi:10.1002/2013WR013764 (2013).

4    Zeng, Y., Su, Z., Wan, L. & Wen, J. A simulation analysis of the advective effect on evaporation using a two-phase heat and mass flow model. *Water Resources Research* **47**, doi:10.1029/2011WR010701 (2011).

5    Zeng, Y., Su, Z., Wan, L. & Wen, J. Numerical analysis of air-water-heat flow in unsaturated soil: Is it necessary to consider airflow in land surface models? *Journal of Geophysical Research Atmospheres* **116**, doi:10.1029/2011JD015835 (2011).

6    Zeng, Y. *et al.* Diurnal pattern of the drying front in desert and its application for determining the effective infiltration. *Hydrol. Earth Syst. Sci.* **13**, 703-714, doi:10.5194/hess-13-703-2009 (2009).

7    Zeng, Y. J. *et al.* Diurnal soil water dynamics in the shallow vadose zone (field site of China University of Geosciences, China). *Environmental Geology* **58**, 11-23, doi:10.1007/s00254-008-1485-8 (2009).

8    Yu, L., Zeng, Y. & Su, Z. Understanding the mass, momentum, and energy transfer in the frozen soil with three levels of model complexities. *Hydrol. Earth Syst. Sci.* **24**, 4813-4830, doi:10.5194/hess-24-4813-2020 (2020).

9    Yu, L., Zeng, Y., Wen, J. & Su, Z. Liquid-Vapor-Air Flow in the Frozen Soil. *Journal of Geophysical Research: Atmospheres* **123**, 7393-7415, doi:10.1029/2018JD028502 (2018).

---

## Author Response (AR2)

**Responses to the Comments by Reviewers**

We thank Professor Bob Su and an anonymous reviewer for their constructive comments. The manuscript has been significantly improved by addressing the comments. The following is our point-to-point responses to their comments.

*1. Line 44-49. The author responds that*

*"The condensate of the soil layer in the semi-arid area is indeed an important source, but the groundwater level in this area is about 7 meters deep, so the groundwater replenishment on soil moisture for shallow soil layers less than 2.2 m deep is essentially negligible. Therefore, the source of condensate at a depth of 2.2 meters comes solely from precipitation, and we can still use the water balance principle to calculate the distribution of precipitation-induced infiltration in each soil layer."*

*First of all, as I commented, even the groundwater is very deep down to hundreds of meters, you still can find continuous upwards vapor flow (either driven by thermal gradient or soil matric potential gradient)1,2. This is essentially important source of water in desert area like your study site. There are also relevant studies implemented over the Badain Jaran Desert, indicating the important role of vapor transport in the sand, and how it affects infiltration and land surface evaporation3-7. Furthermore, the vapor transport is also important for freeze-thaw cycles in soil8,9. As such, the statement of "the source of condensation at a depth of 2.2 meters comes solely from precipitation" cannot be hold.*

*To this reviewer, it is ok you can use simplified equation to derive how much water leave the root zone, and how much infiltrated by precipitation etc.. On the other hand, the total ignorance on the importance of other mechanisms will let readers (incl. this reviewer) to question the scientific rigorousness on this study.*

**Response:** We thank this reviewer for carefully documented the importance of airflow and vapor transport issues. We have created a new section 4.2 to address the limitations and future works. See line 602-646. We have also pointed out this in the conclusion part (see line 683-684).

Specifically, we point out that airflow and vapor transport has not been considered in this experimental investigation and it should be incorporated in future studies because of two considerations.

Firstly, it has been reported that airflow may play an important role for mass and energy transport in arid and semi-arid areas (e.g., Scanlon, 1994; Scanlon and Milly, 1994; Zeng et al., 2009a, 2009b; Zeng et al., 2011a, 2011b; Yu et al., 2018, Yu et al., 2020), this is particuarly true when discussing evaporation process. For instance, Zeng et al. (2011a) has established a one-dimensional (vertical) two-phase heat and mass flow model to explain the field measurements of soil moisture content and temperature in the Badain Jaran Desert of China for both low- and high-permeability soils. They reported that the evaporation was underestimated when the airflow was neglected and such underestimations were more evident in the low-permeability soil (6.4%) as in the high-permeability soil (8.85%). Zeng et al. (2011a) concluded that such underestimations of evaporation were mainly caused by underetimation of isothermal hydraulic conductivity by neglecting airflow. Mohanty and Yang (2013) agreed with Zeng et al. (2011a)

that the underestimation of evaporation was caused by underetimation of isothermal hydraulic conductivity, but they diagreed with Zeng et al. (2011a) that negligience of airflow was responsible for the underestimation of isothermal hydraulic conductivity. The critical comment made by Mohanty and Yang (2013) was disputed by Zeng and Su (2013) who upheld the conclusions of Zeng et al. (2011a), but at the same time Zeng and Su (2013) recognized that some other mechanisms such as adsoption of component of the soil water retention (which has been pointed out by Mohanty and Yang (2013)) can be important and should be included in addition to diffusion, advection, and dispersion when discussing the balance equations of water (liquid and vapor), dry air, and heat. In summary, there is a general consensus that airflow is relevant when discussing the mass and energy transport in the unsaturated zone, particularly near the land-atmospheric surface. However, it is still not fully understood to what degree the airflow has contributed to the land-atmosphere interaction.

Secondly, this study mostly concerns liquid water movement below the shallow soil zones (like 2.2 m below ground surface) with a water table as deep as 7 meters in an semi-arid region. How important is vapor transport in the study site is an open question that should be answered when new experiments are conducted in the future. It is speculated that even for ground watertable as deep as hundres of meters, continuous upward vapor transport either driven by thermal gradient, soil matrix potential, or diffusion and dispersion processes may still exist and can be important source of water in desert area like the site of this study (Scanlon, 1994; Scanlon and Milly, 1994). In the past decades, relevant studies in Badain Jaran Desert of China have indicated that vapor transport has played an important role in regulating infiltration and land surface evaporation (Zeng et al., 2009a, 2009b; Zeng et al., 2011a, 2011b; Zeng and Su, 2013). In addition, the vapor transport is also important for freeze-thaw cycles in the Badain Jaran Desert (Yu et al., 2018, Yu et al., 2020). In summary, further research is needed to quantify the importance of airflow and vapor transport and source of condensation at the study site.

2. Line 108. "the amount of vapor transmission is small, so it is not considered"

Over moist region, it is ok to neglect this vapor transport mechanism. However, in arid and semi-arid area, the vapor transport is dominant, or at least the same important as the liquid flow in the soil. See my point 1 and relevant literatures.

**Response**: Please see reply to comment #1.

3. Line 109 "It is impossible for it to return to atmosphere … and it can only keep going down to recharge the deep soil"

This is a very rough and not necessarily correct statement. As here, the author only assume gravitational potential and ignore the soil water flow driven by soil matric potential gradient.

Although the author assumed that 60cm is the capillary rise, as such, any water below 180cm is essentially kept in the soil. This assumption is very strong, and lacking rigorous physically-base proof. Furthermore, in this site, you still have a thick unsaturated zone between the bottom of

1.8m to the groundwater table 5-7m. Such assumption of all soil water within this thick unsaturated zone is purely driven by gravity is essentially not correct.

**Response**: Please see reply to comment #1.

[revised manuscript text omitted]